# Late Split-Application with Reduced Nitrogen Fertilizer Increases Yield by Mediating Source–Sink Relations during the Grain Filling Stage in Summer Maize

**DOI:** 10.3390/plants12030625

**Published:** 2023-01-31

**Authors:** Tao Deng, Jia-Hui Wang, Zhen Gao, Si Shen, Xiao-Gui Liang, Xue Zhao, Xian-Min Chen, Gong Wu, Xin Wang, Shun-Li Zhou

**Affiliations:** 1College of Agronomy and Biotechnology, China Agricultural University, Beijing 100193, China; 2Innovation Center of Agricultural Technology for Lowland Plain of Hebei, Wuqiao 061802, China

**Keywords:** late split application of N, summer maize, yield, source–sink mediation, N uptake and utilization, limited N fertilizer input

## Abstract

In the North China Plain, the excessive application of nitrogen (N) fertilizer for ensuring high yield and a single application at sowing for simplifying management in farmer practice lead to low N use efficiency and environmental risk in maize (*Zea mays* L.) production. However, it is unclear whether and how late split application with a lower level of N fertilizer influences maize yield. To address this question, a two-year field experiment was conducted with two commercial maize cultivars (Zhengdan 958 and Denghai 605) using a lower level of N input (180 kg ha^−1^) by setting up single application at sowing and split application at sowing and later stages (V12, R1, and R2) with four different ratios, respectively. The maize yield with split-applied 180 kg ha^−1^ N did not decrease compared to the average yield with 240 kg ha^−1^ N input in farmer practice, while it increased by 6.7% to 11.5% in the four N split-application treatments compared with that of the single-application control. Morphological and physiological analyses demonstrated that late split application of N (i) increased the net photosynthetic rate and chlorophyll content and thus promoted the photosynthetic efficiency during the reproductive stages; (ii) promoted the sink capacity via improved kernel number, endosperm cells division, and grain-filling rate; and (iii) increased the final N content and N efficiency in the plant. Therefore, we propose that late split application of N could reduce N fertilizer input and coordinately improve N efficiency and grain yield in summer maize production, which are likely achieved by optimizing the source–sink relations during the grain-filling stage.

## 1. Introduction

For a long time, the productivity increase of cereal crops in China has relied on excessive application of fertilizer, which has become a major barrier for sustainable development of agriculture [1,2]. Actually, only less than half of the fertilizers could be utilized by crops [3,4], and the accumulation of fertilizer nitrogen (N) has caused serious pollution to the atmosphere, soil, and water resources [5,6]. The Ministry of Agriculture and Rural Affairs of China clearly stated that it was necessary to reduce the use of fertilizers and pesticides in agricultural production (http://www.moa.gov.cn/, 2015, accessed on 27 December 2022).

Maize (*Zea mays* L.) has the largest planting area in China (National Bureau of Statistics of China, 2019), and N fertilizer rates of maize used by farmers are usually about 240 kg ha^−1^ [7,8]. To achieve a super-high yield of maize, over 400 kg ha^−1^ of N fertilizer input was reported in some areas [2,7]. In fact, previous studies have demonstrated that less N input, for instance, 180 kg ha^−1^ N, could satisfy the N demand and achieve a high yield of summer maize in China [4,9]. Additionally, the methods of N fertilizer application are improper in some areas. Split application at sowing and at the V12 growth stage is recommended in maize production. However, farmers prefer to simplify the fertilization process by applying all the fertilizer at sowing in practice. This approach induces a series of consequences, including increased denitrification, leaching, volatilization, surface runoff, and thus low N use efficiency (NUE) [10,11,12]. In fact, rational allocation of fertilizer at some critical growth periods of maize could be an efficient way to ensure high yield with limited N input. For instance, partial N fertilizer that is applied later can meet the N requirement from the maize plant at later growth stage, representing a synchronization in N application with the crop growth process [13,14,15]. By this approach, massive N loss caused by seasonal rainfall could be avoided, and the synchronization in nutrients supplying with the demand of crop growth could be facilitated [16]; as a consequence, NUE and N recovery efficiency are both improved [15,17,18].

The effects of a split application of N fertilizer on maize yield are relevant to many factors, including maize genotype, environment conditions, and the time for N split application [17,19,20]; therefore, its effects are usually difficult to accurately clarify. In previous studies, maize yield did not decrease when N fertilizer was put off from the sowing stage to the V10 or V11 stage but decreased when N fertilizer was delayed to the tasseling and silking stages [16,21,22]. Another study reported that yield was increased by 7.1–14.2% when the application of N fertilizer was delayed to stages V6, V10, and 10 days after flowering [23]. In accordance with these findings, a recent simulation by a meta-analysis predicted that a split application of N could increase maize yield by increasing post-silking N uptake with N input varying between 0 and 315 kg N ha^−1^ in different countries [24]. Using the ^15^N labelling approach, the dynamic of N accumulation following maize growth was investigated, and it was found that up to 40–50% of the N in mature grains is taken up from the soil after silking [25,26]. Thus, the N from the fertilizer split-applied after silking would be more easily distributed to the kernels [27]. Although evidence supporting the benefit of a split N application on maize yield is growing, some opposite opinions and cases also exist. It is possible that inappropriate N distribution causes a N deficit at the vegetative stage and thus hinders plant growth [19,28,29]. Therefore, ensuring vegetative growth as well as post-silking growth through a suitable split N application is critical for improving NUE and maize yield.

However, the current understanding of split applications of N at the silking stage, as well as its influence on grain set and grain-filling processes, are still limited in maize. At the grain set stage, several key physiological processes, including pollination, fertilization, and endosperm cell division are accomplished, and thus the grain number as well as yield potential are largely determined [30,31,32]. Pollination failure and environmental stresses occurring at the set stage usually induce grain abortion and subsequent grain number losses [33,34]. Meanwhile, during the grain-filling process, the final kernel weight and the accumulations of starch and protein in the endosperm and embryo are determined [35]. Especially in leaves, the main source, N plays an important role in the synthesis of chloroplasts and the maintenance of enzyme activity related to photosynthesis at the late stages [36,37]. Another study has shown that N fertilizer can increase the N content in leaves and enhance the photosynthetic capacity and assimilate supply during the grain-filling period [37]. Given the decisive role of source–sink relations in determining grain yield, the manner in which a split application of N fertilizers impacts source–sink performance during grain development and thus yield formation requires further study.

Many studies on split applications of N fertilizer, including parts of those mentioned above, were conducted under sufficient N input conditions [23,24,38]. With limited total N input, how the delayed N application influences yield formation in maize is unclear. Our earlier study indicated that 180 kg ha^−1^ of N fertilizer could satisfy the N demand and achieve high yield in summer maize in the North China Plain (NCP) [9], where 240 kg ha^−1^ of N is currently adopted [7,8]. Based on the previous findings, the current study aimed to (i) explore the effects of different patterns of late split applications of N on source–sink performance, N uptake, and grain development dynamic with a total amount of 180 kg ha^−1^ of N fertilizer; (ii) clarify the mechanism of a late split application of N with limited N input affecting summer maize yield formation from the source–sink relations; (iii) and find a suitable N split application pattern to maximize maize yield with reduced N fertilizer input.

## 2. Materials and Methods

### 2.1. Experimental Site

This experiment was conducted at the Wuqiao Experimental Station of China Agricultural University (Hebei Province, China, 37.4° N, 116.3° E) in 2016 and 2019. Average monthly rainfall and temperature during the maize growing season at the Wuqiao Experimental Station are shown in Figure 1. In both years, the previous crop was winter wheat. Before sowing, 0–40 cm-deep soil was roughly measured to contain 10.21 g kg^−1^ of organic matter, 0.9 g kg^−1^ of total N, 16.83 mg kg^−1^ of available phosphorus, and 89.84 mg kg^−1^ of exchangeable potassium.

### 2.2. Experimental Design

In this experiment, four treatments of late split N application based on different application times and ratios of N input were established, while a single application at sowing served as the control. The total amount of N fertilizer for all late split-application treatments and the control was 180 kg ha^−1^. Detailed information of the N treatments is provided in Table 1. Two maize hybrids “Zhengdan 958” (ZD958) and “Denghai 605” (DH605), which are widely planted in China, were used. DH605 has larger ears, a slightly later growth period, and a longer grain-filling duration compared with those of ZD958. A split-plot design was adopted with N fertilizer treatments as the main plot and the two hybrids as the subplots. The subplots were randomized within the main plot. There were three replications per treatment, and each plot was 60 m^2^ with 10 rows in width (0.6 m per row) and 10 m in length.

Forty percent of all late split-application treatments, as well as 100% of the control, of the total amount of N fertilizer was applied on the row side at the sowing stage along with 138.5 kg ha^−1^ of P_2_O_5_, 113 kg ha^−1^ of K_2_O, and 15 kg ha^−1^ of ZnSO_4_. The remaining 60% N for late split-application treatments was applied at the 12-leaf unfolding period (V12), the silking stage (R1), and 15 days after silking (R2) by drip irrigation, respectively (Table 1). The amount of water used for every fertigation was 45 m^3^ ha^−1^, including the control. Before applying the fertilizer, the drip irrigation equipment was adjusted with clean water to ensure that the water rate at both ends of each drip irrigation belt was the same. After each fertilization, the drip irrigation equipment was washed with water for a period of time to ensure that the fertilizer was fully delivered.

The two maize hybrids were planted at a density of 75,000 plants ha^−1^. In 2016, the experimental plants were sown on the 13rd June and harvested on the 27th September. In 2019, dates for sowing and harvest were the 19th June and the 16th October, respectively. The weeds were adequately controlled, and there were no visible water stress or pests throughout the growing season of both years.

### 2.3. Sampling and Measurements

#### 2.3.1. Photosynthetic Rate

The net photosynthetic rate (Pn) was measured using a photosynthetic measurement system (LI-6400, LI-COR Biosciences, Lincoln, Nebraska, FL, USA). Light quantum flux was 1200 μmol m^−2^ s^−1^, and the leaf chamber temperature was 25 °C. Measurements were conducted from 11:00 to 13:00 on a sunny day. Four plants with the same growth state and leaf orientation were selected for each treatment. The Pn in the middle position of the ear leaf was measured. In 2016, the measurement time was at 25 days after silking (25 DAS), and it was at 35 DAS in 2019 due to insufficient sunlight.

#### 2.3.2. Chlorophyll Content

From silking to harvesting, the ear leaves from four plants were obtained from each treatment every 10 d. The middle parts of the leaves were cut and collected from both sides of the veins. A portion (0.15 g) of the sampled leaf was weighed and then immersed in 10 mL of 95% ethanol for 48 h to completely dissolve the chlorophyll. The extract was diluted 10 times by 95% ethanol. The absorbance of the mixture was measured at 649 and 665 nm with a spectrophotometer (UV-3200, Mapada Instruments, Shanghai, China), and chlorophyll content was calculated as described by Li [39]:

Chlorophyll a = 13.95A_665_ − 6.88A_649_;

Chlorophyll b = 24.96A_649_ − 7.32A_665_;

Chlorophyll concentration (mg/L) = Chlorophyll a + Chlorophyll b;

Chlorophyll content (mg/g) = (chlorophyll concentration × volume of extraction solution × dilution factor)/fresh weight of sample.

#### 2.3.3. Determination of Endosperm Cells and Starch Granules

At 15 DAS, three to four typical kernels in the middle ear from four independent plants in each plot center were collected. Determination of endosperm cells was performed as described by He et al. [40]. Briefly, four collected kernels from each treatment were fixed with a Carnoy solution (anhydrous ethanol:glacial acetic acid:chloroform = 9:3:1), vacuum-infiltrated for 48 h, and stored in a 70% ethanol solution at 4 °C. Kernels were dehydrated with ethanol, clarified with xylene, embedded in paraffin, and stained with hematoxylin-eosin. A slice of about 10 μm thickness in the middle of the kernel was cut and observed with a microscope (Olympus BX51, Olympus China Co., Ltd., Beijing, China).

Determination of starch granules was performed as described by Wang et al. [41]. Briefly, about 0.6 μm of thickness in the middle of the collected kernel from each treatment was sliced. The cross section of the kernel was fixed with 4% glutaraldehyde before washing with a phosphate buffer. Then, it was fixed with 1% osmium acid, dehydrated with ethanol gradient, soaked, and embedded with Epon-812 (epoxy resin). The slice was placed on a glass slide and heated to 60 °C. After drying, it was stained with a basic magenta-methylene blue staining solution at 60 °C and observed with a microscope (Olympus BX51)

#### 2.3.4. Grain-Filling Dynamics

At silking, plants with the same growth status (plant height, stem thickness, ear height, and ear size) were marked in each treatment. At 5, 10, 15, 20, 30, 40, and 50 DAS, three ears of similar size were measured for each treatment. Two rows of kernels for each ear were counted, and the fresh weight of the kernel was measured. The kernel volume was determined using the drainage method, and the dry weight of the kernel was measured after drying.

The grain-filling process was fitted using a logistic growth equation as described by Wang et al. [42]: *W = A/(*1 *+ Be^−Ct^)*. The filling phase was calculated from the derivative of the equation, which was different from the physiology phase. The ending date for grain-filling of the early stage was *t*_1_
*= (lnB −* 1.317*)/C*, the deadline for the middle grain-filling stage was *t*_2_
*= (lnB +* 1.317*)/C*, and the ending date of grain-filling for the late stage was *t*_3_
*= (lnB +* 4.59512*)/C*. The durations of the early, middle, and late stages were *T*_1_
*= t*_1_, *T*_2_
*= t*_2_
*− t*_1_*,* and *T*_3_
*= t*_3_
*− t*_2_, respectively. The rates of the early, middle, and late stages were *V*_1_
*= (W*_1_
*− W*_0_*)/T*_1_*, V*_2_
*= (W*_2_
*− W*_1_*)/T_2_*_,_ and *V*_3_
*= (W*_3_
*− W*_2_*)/T*_3_, respectively, where W is the grain weight (mg), t is the time after silking (d), V is the rate of grain filling (mg kernel^−1^ d^−1^), A is the maximum grain weight (mg), and B and C are coefficients determined by the regression.

#### 2.3.5. Determination of N Content and N Traits

The Kjeldahl method [43] was used to determine the total N content in each part of the plant at the R1 and R6 stages. The related parameters and calculation formulas are as follows:

N harvest index (%) = grain N at R6/total plant N at R6 × 100;

Remobilized N (g) = (R1 leaf N + R1 stem N) − (R6 leaf N + R6 stem N);

N fertilizer partial productivity = yield/total N fertilizer input.

#### 2.3.6. Grain Yield

Maize plants of two rows with 5 m length in the middle region of each plot were harvested, and the total fresh weight of all ears was measured. Based on the average ear weight and proportion of ear size, 20 ears were selected for further investigation. Agronomic traits, including ear length, ear thickness, bald tip length, ear rows, kernels per row, 100-grain weight, and water content were examined, and actual yield was calculated based on 14% moisture content.

#### 2.3.7. Data Analysis

Significance was assessed with SPSS 25 (SPSS Inc., Chicago, IL, USA) and Excel 2016 using one-way analysis of variance (ANOVA) with Duncan’s test. A *p*-value of <0.05 was considered as statistically significant.

## 3. Results

### 3.1. Yield and Yield Components

With limited N fertilizer input, the late split application of N increased the maize yield compared with that with a single fertilization at sowing (Table 2). In 2016, the yield of ZD958 in N3 and of DH605 in N2 and N3 was significantly increased by 14.4%, 19.2%, and 13.5% compared with that under CK, respectively. In 2019, the yield of ZD958 in N3 and of DH605 in N1 and N4 was significantly increased by 14.0%, 16.0%, and 13.0% compared with that with CK, respectively. Within the four different patterns of split N application, the N3 pattern with a ratio of sowing/12-leaf unfolding period/silking stage = 4:2:4 gave the highest yield, increasing the average yields of ZD958 and DH605 by 14.2% (12.05 t ha^−1^) and 8.3% (11.05 t ha^−1^), respectively. Two years of experimental data indicated that the yield increase under the late split application of N was mainly due to increased kernel number. In ZD958, the kernel number of the N1 to N4 treatments was increased by 5.7%, 4.5%, 9%, and 8.2% compared with that with CK, respectively. In DH605, the kernel number was increased by 6%, 5.6%, 7.2%, and 8.6% compared with that with CK, respectively. The increase in kernel weight in some of the N treatments also slightly contributed to the yield.

### 3.2. Source Performance

The late split application of N increased the net photosynthetic rate of the maize after silking in both varieties compared with that with CK (Figure 2). There were also significant differences among the N3, N4, and N1 treatments. The results indicated that the application of N fertilizer during the silking stage or later further increased the net photosynthetic rate after the silking stage, on the basis of partial N fertilizer application in V12.

The late split application of N increased the chlorophyll content of the maize after silking (Figure 3). The average chlorophyll content in the treatments was higher than that with CK throughout the grain-filling process, especially during the middle stages. In multiple periods of grain filling, the N3 and N4 treatments increased the chlorophyll content significantly compared with that with CK, and the difference was more obvious in the middle and late stages. This suggested that the application of N fertilizer during the silking stage or later could enhance the chlorophyll content in the middle and late stages of grain filling.

### 3.3. Sink Performance

In order to study the effect of a late split N application on sink establishment and development, endosperm cells and starch granules in the middle kernel of the ear at 15 DAS were observed. It was found that the late split application of N could significantly increase the endosperm cell number (Figure 4). Compared with that with CK, the N1 to N4 treatments of ZD958 increased the endosperm cell number by 37.6%, 38.5%, 46.0%, and 31.3%, respectively, while in DH605 it was increased by 35.7%, 46.1%, 49.6%, and 16.7%, respectively. Meanwhile, an obvious increase of the number of starch granules in endosperm cells was observed in late split N application treatments (Figure 5). The N3 treatment greatly increased the number of endosperm cells and starch granules in both varieties, indicating that N fertilizer applied at the silking stage played a pivotal role in expanding the potential sink capacity.

A logistic equation was used to fit the grain-filling process and determine the effects of different treatments on important grain-filling parameters (Table 3). It indicated that the late split application of N did not affect the duration of the grain-filling process but significantly increased the filling rate of the middle stage (V2). In 2016, compared with that with CK, split N application increased grain filling in the V2 stage by an average of 6.8% and 9.8% in ZD958 and DH605, respectively. In 2019, it increased gran filling by an average of 11.3% in ZD958 and 9.3% in DH605. In addition, the late split application of N had a certain increase on the filling rate of the late stage (V3), whereas the two-year data did not show any strong patterns, which may be related to a sampling error. Overall, the rate of each filling stage (V1, V2, and V3) in the N3 treatment was significantly improved compared with that with CK, thus performing better than the other treatments.

### 3.4. N Uptake and Utilization

The late split application of N significantly increased plant N accumulation in the mature stage (Figure 6). According to the two-year data, the N accumulation of ZD958 and DH605 increased by an average of 11.2% and 9.3% in the N3 treatment compared with that with CK. Meanwhile, an average increase of 12.3% and 13.6% was observed in ZD958 and DH605 in the N4 treatment. It indicated that 20% of the N fertilizer allocated at 15 DAS could be better absorbed by the plants. The increase of total N accumulation was mainly achieved by an increase of the leaves, which increased by 13% compared with that with CK. The distribution of N in the stems and leaves of mature plants was about 14% and 36%, respectively; the kernels accounted for about 30%; and the rest was present in the cobs and husks of the ear. Therefore, the late split application of N significantly increased total N accumulation at the maturity stage and promoted N distribution to the kernels. Notably, N3 and N4 fertilization increased the kernels’ N content by approximately 7% and 8%, respectively, performing the best of all treatments.

At the R1 stage, there was no significant difference of N accumulation between all and 60% of N input before silking in the whole plant or in any organ (Figure 7). This demonstrated that 60% of N fertilizer input before silking could satisfy the N requirements during the vegetative growth stage, leading to an excess of 40% fertilizer waste.

In terms of N efficiency (Table 4), the partial productivity of N fertilizer has been greatly improved compared with that with CK, and the increase in the N harvest index and remobilized N was not obvious. Taken together, the N3 and N4 treatments performed better than the other treatments in terms of N uptake and utilization characteristics.

## 4. Discussion

### 4.1. Late Split Application of N Coordinately Improves Yield and N Efficiency in Maize under Limited N Input

N efficiency is closely related to yield and could reflect the yield output of maize, and its parameters generally include NUE, N recovery efficiency, remobilized N, N fertilizer partial productivity, and the N harvest index [15,24]. In the current study, the partial productivity of N fertilization under the N3 treatment was significantly higher than that with CK, while remobilized N increased slightly (Table 4). The increase of remobilized N meant that the transfer of N from the vegetative organ of the silking stage to maize kernels was increased (Table 4), and it was not difficult to find that N in kernels increased at the maturity stage (Figure 6). Therefore, we believe that the late split application of N under limited N input could promote the absorption of N in plants, which is conducive to transferring N into grains and thus improving N efficiency, in line with previous conclusions [17,18,27].

However, the impact of a split application of N on yield is sometimes inconsistent. Some split-application N experiments, which applied the fertilizer at the V12 or even R1 stages, resulted in a reduction in yield [16,20,21,38], as it ignores the N demand for crop growth and development in the early stages, which may cause irreparable damage to the crop and affect the absorption of N fertilizer in the later stages. Additionally, a study found that there was an increase in N fertilizer recovery rate but not in yield when N fertilization was moved to the V3 and V12 stages under high N fertilizer input levels [17], implying that this approach might have satisfied the N fertilizer demand in the vegetative growth stage but neglected the N fertilizer input in the reproduction growth stage. In the current study, the total input of N fertilizer was reduced to 180 kg ha^−1^, 40% of which was applied at sowing, and the N fertilizer input ratio was increased during the silking stage or later. Under the late split application of N, the kernel number, kernel weight, and maize yield increased compared with that under CK; especially the yield and kernel number increased significantly when 40% of the N fertilizer was applied at the silking or later stages compared with that under CK (Table 2).

On the other hand, we collected and investigated the experimental data of maize yield involved in the N input amount adopted by farmers since 2015 to 2018 at the NCP experimental site (located at Hebei, Shandong, Henan, and Shanxi provinces, China) from published articles. In total, 68 samples were examined, and each sample represented N fertilizer input and yield level with a specific variety in different regions. We calculated the average value of the samples and found that the average yield was 10.1 t ha^−1^ and the average N input was 240 kg ha^−1^ (Appendix A). Significantly, lower N input of 180 kg ha^−1^ did not reduce maize yield (Table 2). Therefore, the late split application of N could reduce N fertilizer input and achieve a coordinated increase in yield and N efficiency. The results of this study provide a theoretical basis for further reducing N fertilizer input, improving N efficiency, and increasing maize yield by optimizing fertilization measures. Additionally, it provides new approaches for the sustainable development of agriculture in China.

### 4.2. Late Split Application of N with Limited N Input Promotes Sink Development and Mediates Source–Sink Relations in the Later Growth Stages of Maize

The source–sink relationship reflects the accumulation of photosynthetic materials and has been an important indicator for evaluating high crop yields. “Source” usually refers to the producer of photosynthetic substances, while “sink” refers to the importer of photosynthetic substances [44]. Whether yield is limited by the source or the sink has been consistently debated; nowadays, an increasing numbers of studies have demonstrated that both of them jointly limit plant growth [45,46]. However, in many experiments with a split application of N, the influences of different N distributions on the performances of the source and the sink have been largely neglected.

It is generally believed that source performance is reflected by the photosynthetic rate and chlorophyll content in the leaf, which represent the rate and duration of the production of photosynthetic substances. In this study, delaying N fertilization to the silking stage significantly enhanced source performance by increasing the photosynthetic rate and chlorophyll content in maize ear leaves after the silking stage (Figure 2 and Figure 3), effectively preventing the maize from precocious completion of the filling process and early leaf senescence caused by insufficient N content at the late stage.

As for sink performance, it can be reflected by the kernel number, kernel weight, and grain-filling rate, which represent the maximum number of photo-contracts accepted and the rate of the acceptance process. Grain setting or sink establishment is determined by endosperm cell division and starch granule formation, which generally occurs within 15 DAS. Some studies have shown that the grain-filling rate and final kernel weight have a positive correlation with the number of endosperm cells [32,47]. Additionally, during the process of grain setting, hormone stimulation can increase the number of endosperm cells, [48] and N fertilizer could boost cytokinin content [49]. In this study, the late split application of N obviously increased the kernel number (Table 2) and promoted endosperm cell division during the grain setting stage to improve the sink potential (Figure 4). Afterwards, it enhanced the filling rate to increase kernel weight (Table 3).

Therefore, splitting N fertilization to the silking stage coordinately improved the source performance and increased the sink capacity in maize, providing a foundation for increasing grain production.

### 4.3. Split N Input at the Silking Stage Meets Maize N Demand of the Late Stage and Increases Economic Benefit under Limited N Input

In general, the N uptake peak in maize is around the silking stage (V10–R2) [50,51] therefore, sufficient fertilizer input at the silking stage is essential to obtaining the highest N uptake efficiency. Furthermore, another study found that maize kernel development is affected by the growth status around the silking stage, and enhancing the plant growth rate can increase the potential sink capacity [52]. However, maize plants with a taller height at the silking stage will cause inconvenience of fertilization and additional labor demand, resulting in a poor adoption of fertilization during this stage in practice. In recent years, with the popularization of drip irrigation equipment and the large-scale application of drones in agriculture, the late split application of N fertilizer has become easier and economically viable. Meanwhile, slowly released fertilizers are another good choice for setting up the released amount at the late stage of the fertilizer to achieve a late split application of N fertilizer.

In this experiment, we found that the application of 60% of N fertilizer at sowing and V12 was sufficient to ensure maize growth and development (Table 2, Figure 7), which was consistent with previous findings [38,53]. To further investigate the effects of N fertilizer input at the silking stage and later on maize growth, the N3 (sowing/V12/R1 = 4:2:4) and N4 treatments (sowing/V12/R1/R2 = 4:2:2:2) were established. Based on these results, the N3 and N4 treatments did not vary in their source–sink performance as well as in N uptake and utilization characteristics, while the yield of the N4 treatment was slightly decreased compared with that of N3 (Table 2). However, at the R6 stage, the leaf N content in the N4 treatment was significantly higher than in N3 and CK (Figure 6). N application at the R2 stage in the N4 treatment may have extended the maize growth period, which is not conducive to the subsequent winter wheat planting.

We also gave an economic evaluation of the late split application of N with limited N input based on this experiment. The N fertilizer amount (180 kg ha^−1^) in this experiment decreased by 60 kg ha^−1^ compared with the traditional fertilizer application amount (240 kg ha^−1^), saving around 180 yuan ha^−1^ (the price of N fertilizer was calculated at 3 yuan kg^−1^). The optimal N split-application mode increased the yield by 1.5 t ha^−1^ and brought about 1950 yuan ha^−1^ in income (the price of maize grain was calculated at 1300 yuan t^−1^). The additional cost was mainly in the purchase of drip irrigation equipment; considering that the equipment has a service life of 5–6 years, the average cost was equivalent to 1500 yuan ha^−1^. In total, the technology of split-application N could increase income by 630 yuan ha^−1^.

Collectively, the N late-split-application mode of the N3 treatment (sowing/V12/R1 = 4:2:4) could coordinately improve yield and N efficiency in maize under limited N input and achieve higher incomes. It should be further implemented in the NCP to promote the sustainable development of agriculture with less fertilizer in this region.

## 5. Conclusions

With limited N fertilizer input, late split application of N significantly increased the yield of summer maize in the NCP. The application of 60% of N fertilizer before silking satisfied the N requirements during the vegetative growth stage, and the remaining 40% of N fertilizer was recommended to be applied at the silking stage. This strategy could mediate the source–sink balance by promoting sink development, coupled with enhancing source performance during the grain-filling stage, and coordinately improve yield and N efficiency, displaying better performance than the other three treatments.

## 6. Patents

This section is not mandatory but may be added if there are patents resulting from the work reported in this manuscript.

## Figures and Tables

**Figure 1 plants-12-00625-f001:**
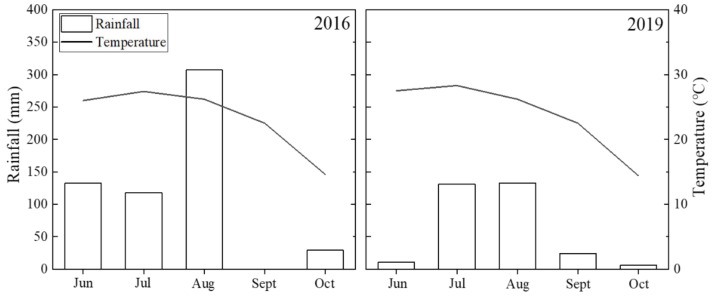
Average monthly rainfall and temperature during maize growing season (1 January to 31 October) in 2016 and 2019.

**Figure 2 plants-12-00625-f002:**
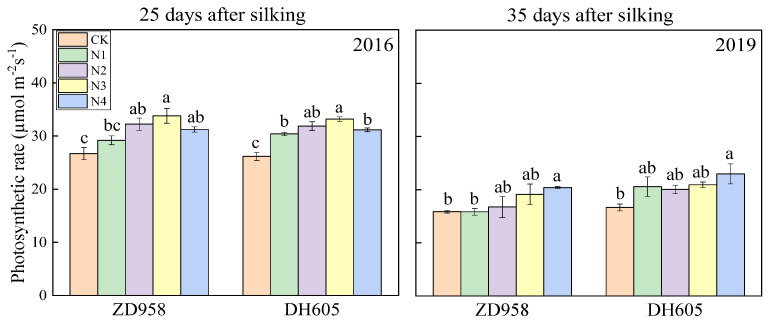
Effects of N treatments on photosynthetic rate of ear leaves in maize in 2016 and 2019. One-way analysis of variance with Duncan’s new multiple range test was conducted to assess the statistical patterns. Different letters indicate a significant difference (*p* < 0.05).

**Figure 3 plants-12-00625-f003:**
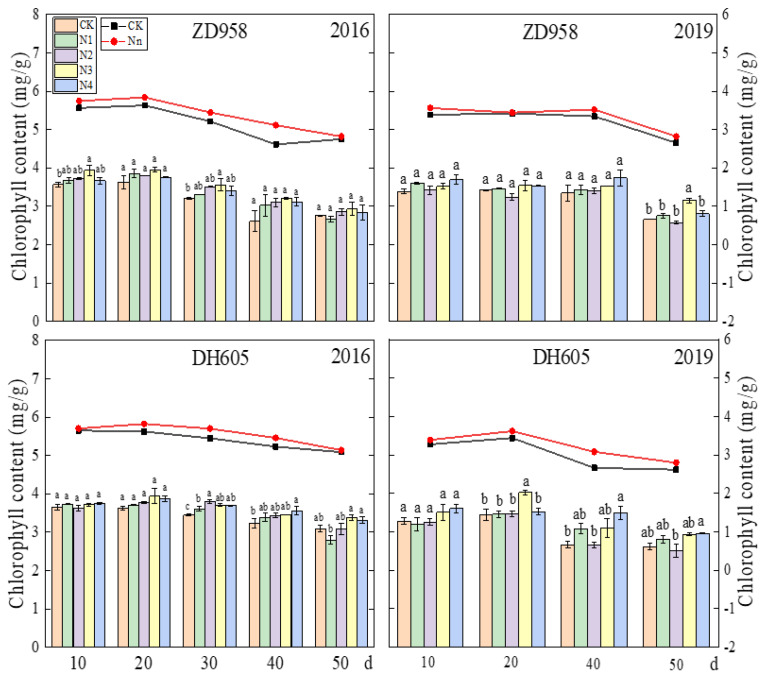
Effects of N treatments on chlorophyll content of ear leaves in maize in 2016 and 2019. The chlorophyll content was measured every 10 days after silking (as shown on the left axis). Black and red lines indicate the chlorophyll content under the control and the late split N treatments (average value), respectively (as shown on the right axis). One-way analysis of variance with Duncan’s new multiple range test was conducted to assess the statistical patterns. Different letters indicate a significant difference (*p* < 0.05).

**Figure 4 plants-12-00625-f004:**
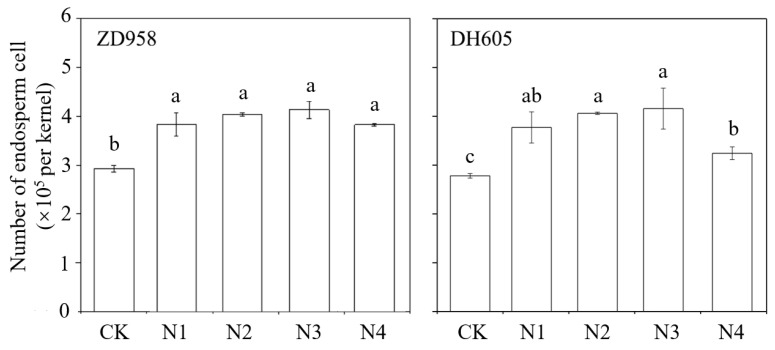
Effects of N treatments on endosperm cell number at 15 days after silking in 2016. One-way analysis of variance with Duncan’s new multiple range test was conducted to assess the statistical patterns. Different letters indicate a significant difference (*p* < 0.05).

**Figure 5 plants-12-00625-f005:**
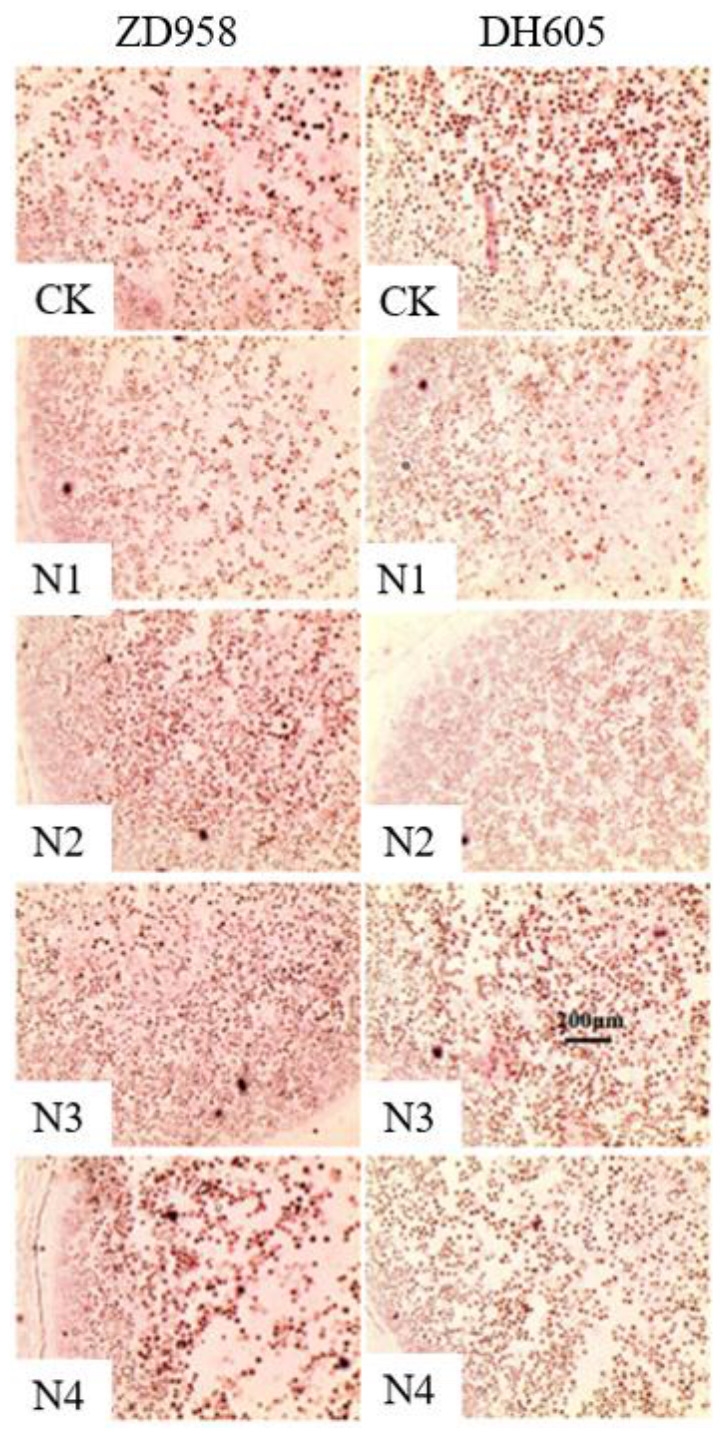
Starch granule in endosperm cell in different treatments at 15 days after silking in 2016. CK means a single application of N at sowing; N1, N2, N3, and N4 indicate four treatments of late split N application based on different application times and ratios of N input. A black dot represents a starch grain, and the scale bar indicates 200 µm.

**Figure 6 plants-12-00625-f006:**
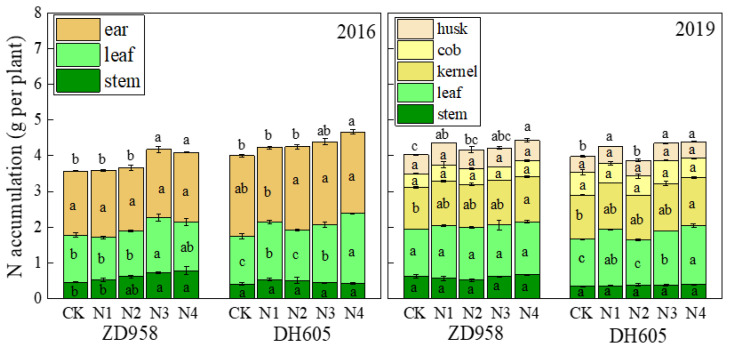
Effects of N treatments on N accumulation in the R6 stage of maize in 2016 and 2019. Total N contents of the leaf and stem were examined, and the ear was further divided into the husk, cob, and kernel in 2019. One-way analysis of variance with Duncan’s new multiple range test was conducted to assess the statistical patterns. Different letters indicate a significant difference (*p* < 0.05).

**Figure 7 plants-12-00625-f007:**
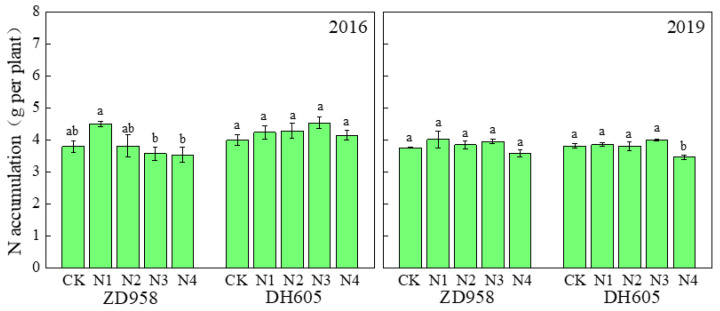
Effects of N treatments on total N accumulation in the R1 stage of maize in 2016 and 2019. Total N contents of the leaf and stem were examined. One-way analysis of variance with Duncan’s new multiple range test was conducted to assess the statistical patterns. Different letters indicate a significant difference (*p* < 0.05).

**Table 1 plants-12-00625-t001:** N fertilizer treatments in 2016 and 2019.

Treatment	Total N Fertilizer Amount (kg ha^−1^)	Application Time and Percentage			
		Sowing	V12	R1	R2
CK	180	100%	0	0	0
N1	180	40%	60%	0	0
N2	180	40%	0	60%	0
N3	180	40%	20%	40%	0
N4	180	40%	20%	20%	20%

V12, twelve-leaves unfolding period; R1, silking stage; R2, 15 days after silking. CK means a single application of N at sowing; N1, N2, N3, and N4 indicate four treatments of late split N application based on different application times and ratios of N input.

**Table 2 plants-12-00625-t002:** Effects of N treatments on maize yield and yield components.

Hybrid	Treatment	2016					2019		
		Ears per m^2^	Kernels per Ear	100-Kernels Weight (g)	Grain Yield (t ha^−1^)	Ears per m^2^	Kernels per Ear	100-Kernels Weight (g)	Grain Yield (t ha^−1^)
ZD958	CK	7.5 a	444 b	27.9 a	10.4 b	7.5 a	499 a	26.7 b	10.7 bc
	N1	7.5 a	476 a	28.1 a	10.6 ab	7.8 a	520 a	26.2 b	11.5 ab
	N2	7.6 a	490 a	28.3 a	11.2 ab	7.8 a	492 a	26.4 b	10.4 c
	N3	7.6 a	493 a	28.6 a	11.9 a	7.6 a	534 a	27.4 a	12.2 a
	N4	7.3 a	487 a	28.5 a	10.8 ab	7.5 a	532 a	26.4 b	11.4 ab
DH605	CK	7.5 a	435 c	31.2 b	10.4 b	7.3 a	478 b	26.1 b	10.0 b
	N1	7.6 a	466 ab	31.8 ab	11.3 ab	7.5 a	501 ab	27.1 a	11.3 a
	N2	7.5 a	483 a	32.0 ab	12.4 a	7.7 a	479 b	26.7 ab	10.2 b
	N3	7.5 a	478 a	32.2 a	11.8 a	7.7 a	499 ab	26.3 ab	10.3 b
	N4	7.6 a	450 bc	31.9 ab	11.0 ab	7.4 a	544 a	26.4 ab	11.6 a
ANOVA	Year	NS	***	***	NS				
	Hybrid	NS	**	*	NS				
	Treatment	**	***	NS	***				
	Y × H	NS	NS	**	**				
	Y × T	NS	**	*	***				
	H × T	NS	NS	NS	*				
	Y × H × T	NS	NS	NS	NS				

One-way analysis of variance with Duncan’s new multiple range test was conducted to assess the statistical patterns. Different letters indicate a significant difference (*p* < 0.05). NS, not significant; *, **, and *** are significant at 0.05, 0.01, and 0.001, respectively. Y, year; H, hybrid; T, treatment.

**Table 3 plants-12-00625-t003:** Effects of N treatments on grain-filling parameters in summer maize.

Year	Hybrid	Treatment	T1	T2	T3	V1	V2	V3
			(Days)			(mg kernel^−1^ d^−1^)		
2016	ZD958	CK	18.3 b	15.7 b	19.5 b	2.9 b	9.1 c	2.6 c
		N1	18.2 b	16.4 ab	20.5 ab	3.3 ab	9.9 a	2.8 a
		N2	18.4 b	17.4 ab	21.7 ab	3.2 ab	9.4 bc	2.6 bc
		N3	20.2 a	18.8 a	23.4 a	3.4 ab	9.9 a	2.8 a
		N4	18.2 b	17.3 ab	21.6 ab	3.4 a	9.8 ab	2.7 ab
	DH605	CK	20.8 ab	19.0 ab	23.6 ab	3.0 b	8.9 b	2.5 b
		N1	22.1 a	20.5 a	25.5 a	3.3 a	9.8 a	2.8 a
		N2	21.2 ab	20.0 ab	24.9 ab	3.4 a	9.8 ab	2.7 ab
		N3	20.6 ab	18.1 b	22.5 b	3.2 a	9.9 a	2.8 a
		N4	20.2 b	18.8 ab	23.4 ab	3.3 a	9.7 ab	2.7 ab
2019	ZD958	CK	21.4 a	13.3 b	17.8 a	2.3 a	9.0 b	2.5 b
		N1	22.5 a	13.8 a	17.7 a	2.2 a	9.1 b	2.5 b
		N2	22.1 a	13.5 ab	16.8 a	2.3 a	10.2 a	2.9 a
		N3	21.4 a	13.1 b	16.3 a	2.6 a	10.9 a	3.1 a
		N4	22.1 a	13.6 ab	17.3 a	2.3 a	10.1 ab	2.8 ab
	DH605	CK	23.4 a	17.2 a	21.4 a	2.1 a	7.9 b	2.3 a
		N1	24.0 a	18.5 a	23.1 a	2.3 a	8.2 ab	2.3 a
		N2	24.3 a	17.4 a	21.6 a	2.2 a	8.3 ab	2.3 a
		N3	22.6 a	15.6 a	19.5 a	2.4 a	9.3 a	2.6 a
		N4	23.8 a	17.2 a	21.4 a	2.3 a	8.7 ab	2.4 a

T1, T2, and T3 represent the duration of the lag, middle, and late stages, respectively. In addition, V1, V2, and V3 delegate the grain-filling rate of the lag, middle, and late stages, respectively. One-way analysis of variance with Duncan’s new multiple range test was conducted to assess the statistical patterns. Different letters indicate a significant difference (*p* < 0.05).

**Table 4 plants-12-00625-t004:** Effects of N treatments on N harvest index, remobilized N, and N fertilizer partial productivity.

Hybrid	Treatment	2016			2019		
		NHI (%)	R.N (g)	PFPN (kg kg^−1^)	NHI (%)	R.N (g)	PFPN (kg kg^−1^)
ZD958	CK	50.5 ab	2.01 a	58.0 b	51.7 a	1.81 ab	59.4 bc
	N1	52.4 a	1.79 a	59.0 ab	53.2 a	1.73 ab	63.9 ab
	N2	48.5 abc	1.92 a	62.0 ab	52.0 a	1.95 a	57.8 c
	N3	45.9 c	1.54 a	66.0 a	51.2 a	1.94 a	67.8 a
	N4	47.8 bc	1.52 a	60.2 ab	51.5 a	1.46 b	63.3 ab
DH605	CK	56.6 a	2.25 a	58.1 b	58.2 a	2.13 a	55.6 b
	N1	49.5 b	2.33 a	62.5 ab	54.6 bc	1.98 b	62.8 a
	N2	54.9 a	2.48 a	68.9 a	57.4 ab	1.98 b	56.7 b
	N3	52.9 ab	2.62 a	65.6 a	56.6 ab	2.11 a	57.2 b
	N4	49.0 b	1.99 a	61.1 ab	53.4 c	1.34 c	64.4 a

NHI, N harvest index; R.N, remobilized N; PFPN, N fertilizer partial productivity. In 2016, nitrogen content in each part of the ear was not measured at R6. Instead, it was measured after crushing the whole ear. One-way analysis of variance with Duncan’s new multiple range test was conducted to assess the statistical patterns. Different letters indicate a significant difference (*p* < 0.05).

## Data Availability

All data supporting the results of this research are included within the article.

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
