# Peer review of "Late Split-Application with Reduced Nitrogen Fertilizer Increases Yield by Mediating Source–Sink Relations during the Grain Filling Stage in Summer Maize"

_plants, 2023, doi:10.3390/plants12030625_

Round 1

Reviewer 1 Report

This work showed the impact the effects of different patterns of late split-application of N on summer maize yield with a total amount of 180 kg ha-1 N fertilizer. In the study the source-sink performance, N uptake and grain development dynamic were investigated, and the effects of late-split-application on summer maize production were evaluated as well.  The experiment were carried out in two years 2016 and 2019. The Authors present very interesting results. Work is written correctly,  research methods selected correctly, results presented in a clear manner, enough literature included, although it was found a missing of one number in the text of paper. Conclusions are answering for main aims of the study. In the body of manuscript are two questions for answering. A few corrections are needed and are marked at the text of the paper.

After all corrections the paper can be publish at the MDPI Plants Journal

Reviewer 2 Report

A well written abstract, highlighting the results of the 2 year experiments with split application having significantly higher yield compared to single application at sowing.  

The research question of how to reduce N application and not suffer any yield penalty was addressed in the introduction, setting the scene why this study is relevant for Chinese farmers growing maize.  

P2 L47: Typo “recommded” to “recommended”

P2 L52: Suggest changing ‘ delay- applied’ to applied later’

P2 L62:  The word ‘postponement ‘ is confusing here does not make sense in the senstence, use simpler language

P2 L89: Replace considered with Given

It would be useful if the authors end the introduction section with the objectives clearly stated instead of just haphazard sentences. 

A well-written material and methods section but some more details needs to be included for instance, how many times were the leaves measured for photosynthesis, was it on same leaf and over which time period. Every section was quite brief and need more information.

Also, most experiments are conducted in 2 consecutive years, this one was 3 years apart, any particular reason why this delay.

The results could not be compared between the two years as they are conducted at different growth stages, a 10-day difference have huge environmental effect on the results shown. The results presented were very basic again making it difficult to compare across the two years. The data could have been analysed better using cross year analysis despite the different growth stages to see the effect of year on N traits. No results were shown on NUE or any of the N traits mentioned in the methodology section in this section but then shown in discussion! The section was written mostly as a report instead of distilling down the reason why the yield was better with splits N application. Figure 5 is a strength of the manuscript with the images of the starch granules. 

Why is table 4 in discussion and not in results. 

The discussion is written poorly, the authors reported the results instead of discussing the results which was already done in results section. The reasons why the split application resulted in higher yield was not discussed properly nor was it linked to the N traits of the study.  A more comprehensive discussion is needed to make it a stronger manuscript and not a repetition of results section. There is a substantial lack of proper referencing to compare the authors findings.

The conclusion is quite weak, just stating a 60-40 splits at vegetative and silking stage  but does not mentioned the others. 
